# A fast lasso-based method for inferring higher-order interactions

**Kieran Elmes**[1,2], **Astra Heywood**[3], **Zhiyi Huang**[1], **Alex Gavryushkin**[2]*

**1** Department of Computer Science, University of Otago, Dunedin, New Zealand, **2** School of Mathematics and Statistics, University of Canterbury, Christchurch, New Zealand, **3** Department of Biochemistry, University of Otago, Dunedin, New Zealand

* alex@biods.org

## Abstract

Large-scale genotype-phenotype screens provide a wealth of data for identifying molecular alterations associated with a phenotype. Epistatic effects play an important role in such association studies. For example, siRNA perturbation screens can be used to identify combinatorial gene-silencing effects. In bacteria, epistasis has practical consequences in determining antimicrobial resistance as the genetic background of a strain plays an important role in determining resistance. Recently developed tools scale to human exome-wide screens for pairwise interactions, but none to date have included the possibility of three-way interactions. Expanding upon recent state-of-the-art methods, we make a number of improvements to the performance on large-scale data, making consideration of three-way interactions possible. We demonstrate our proposed method, Pint, on both simulated and real data sets, including antibiotic resistance testing and siRNA perturbation screens. Pint outperforms known methods in simulated data, and identifies a number of biologically plausible gene effects in both the antibiotic and siRNA models. For example, we have identified a combination of known tumour suppressor genes that is predicted (using Pint) to cause a significant increase in cell proliferation.

## Author summary

In recent years, large-scale genetic data sets have become available for analysis. These large data sets often stretch the limits of classic computational methods, requiring too much memory or simply taking a prohibitively long time to run. Due to the enormous number of potential interactions, each gene or variation in the data is often modelled on its own, without considering interactions between them. Recently, methods have been developed to solve regression problems that include these interacting effects. Even the fastest of these cannot include three-way interactions, however. We improve upon one such method, developing an approach that is significantly faster than the current state of the art. Moreover, our method scales to three-way interactions among thousands of genes, while avoiding a number of the limitations of previous approaches. We analyse large-scale simulated data, antibiotic resistance, and gene-silencing data sets to demonstrate the accuracy and performance of our approach.

**Data Availability Statement:** Code for the R package Pint is available at github.com/bioDS/Pint. Relevant data and data processing scripts are available at github.com/bioDS/lasso_data_processing.

**Funding:** We acknowledge support from the Royal Society Te Apārangi through a Rutherford Discovery Fellowship (RDF-UOC1702). This work was partially supported by Ministry of Business, Innovation, and Employment of New Zealand through an Endeavour Smart Ideas grant (UOOX1912) and a Data Science Programmes grant (UOAX1932). The funders had no role in study design, data collection and analysis, decision to publish, or preparation of the manuscript.

**Competing interests:** The authors have declared that no competing interests exist.

This is a *PLOS Computational Biology* Methods paper.

## Introduction

Epistatic gene interactions have practical implications for personalised medicine, and synthetic lethal interactions in particular can be used in cancer treatment [1]. Discovering these interactions is currently challenging at a practical scale [2–5], however. In particular, there are no methods able to infer three-way effects. For a given number of genes there are exponentially many potential interactions, complicating computational methods. If we restrict our attention to pairwise effects, it is possible to experimentally knock out particular combinations of genes to determine their combined effect [6]. This approach does not scale to the approximately 200 million pairwise combinations possible among human protein coding genes, however, let alone 1.3 trillion three-way combinations. We instead consider inferring interactions from large-scale genotype-phenotype data. These include mass knockdown screens, in which a large number of genes are simultaneously suppressed, and the resulting phenotypic effects are measured.

We have shown in [5] that a lasso-based approach to inferring interactions from an siRNA perturbation matrix is a feasible method for large-scale pairwise interaction detection. In this additive model, we assume fitness is a linear combination of the effects of each gene, and the effect of every combination of these genes. For the sake of scalability, in [5] we considered only individual and pairwise effects, and assumed gene suppression was strictly binary. The fitness difference (compared to no knockdowns) in each experiment is then the sum of individual and pairwise effects $\sum_i^p \beta_i g_i + \sum_i^p \sum_{j>i}^n \beta_{i,j}(g_i \cdot g_j)$, where $g_i = 1$ if gene $i$ is knocked down, 0 otherwise. With sufficiently many such mass-knockdowns, we can infer pairwise interactions by finding the pairs of genes whose effect is not the sum of the effects of each gene individually.

Neither of the previously tested inference methods for this model, glinternet and xyz, are effective at the genome-scale however. Glinternet suffers from prohibitively long running times (Fitting interactions in an siRNA screen of 1, 000 genes with ten siRNAs per gene takes several days using ten cores on an Opteron 6276), and xyz does not accurately predict effects in our larger simulations. Our aim is to fit a model including all $p \approx 20,000$ human protein-coding genes, with as many as $n = 200,000$ siRNAs. Furthermore, we aim to go beyond pairwise interactions to consider three-way effects.

A recently developed method, WHInter [7], is effective at solving lasso regression on much larger scale data than glinternet. This performance comes as a result of pruning the interactions considered based on the current regularisation parameter, removing interactions that could not possibly have a non-zero effect. Because doing so does not affect the solution of the regression problem, we expect comparable accuracy from WHInter and glinternet. Nonetheless there are a number of areas in which we can improve upon WHInter. In particular WHInter does not make use of multi-core CPUs, and considers only pairwise interactions.

We have developed an R-package, Pint, that is able to perform square-root lasso regression on all pairwise interactions on a one thousand gene screen, using ten thousand siRNAs, in 15 seconds, and all three-way interactions on the same set in under ten minutes. We are also able to find the largest 2, 000 effects, excluding three-way combinations, on a genome-scale data set with 19, 000 genes and 6, 700 siRNAs in half an hour using two eight-core CPUs. This is made possible by taking into account that our input matrix $X$ is both sparse and strictly binary, parallelising the pruning method from [7], and compressing the active set. To allow three-way

interactions, we extend to a two-step pruning method able to rule out both pairwise and three-way interactions. Our package, Pint, is available at github.com/bioDS/pint.

Our method is based on an existing fast algorithm [8], which we adapt for use on binary matrices. We further add parallelised version of the pruning step from [7], expanded to include three-way interactions. We provide a detailed description of our implementation, followed by the scalability analysis, below. We also perform a simulation study to compare our method's scalability with known methods, and analyse two large-scale experimental data sets.

In the first, an siRNA perturbation screen from [9], we search for both individual genes and combinations (pairwise or three-way) that have an effect when simultaneously silenced, stopping after the first 100 effects have been identified. The results include 22 individual, 41 pairwise and 68 three-way effects. We investigate the biological plausibility of the top five effects, and find that three out of five are suppressing genes that could be related to cell survival. One combination in particular involves simultaneously disabling two tumour suppressing genes, and is predicted to cause a significant increase in cell proliferation.

The second data set is composed of genetic variants identified in the intrinsically antibiotic resistant bacteria *Pseudomonas aeruginosa*. *P. aeruginosa* is an opportunistic pathogen found in a variety of environments and is a leading cause of morbidity and mortality in immunocompromised individuals or those with cystic fibrosis [10, 11]. *P. aeruginosa* is known to acquire adaptive antibiotic resistance in response to long term usage of antibiotics associated with chronic infections [12–14]. The genomes included in that data set are from strains that have been isolated from chronic and acute infections as well as environmental samples. The minimum inhibitory concentration for the antibiotic Ciprofloxacin has been used as the phenotypic marker for this data set. Ciprofloxacin belongs to the fluoroquinolone class of bactericidal antibiotics that targets DNA replication and is one of the most widely used antibiotics against *P. aeruginosa* [15].

This set contains over 170, 000 SNVs, too many for our method to include all possible three-way interactions. Three-way interactions can be included if we remove columns with less than 30 entries, reducing the matrix to $\approx 76, 000$ columns. This ignores over half of the SNVs present however, so we instead limit the search to individual and pairwise combinations of variants, and determine the first 50 effects that are discovered. Among these, 13 of the 16 non-synonymous variants were identified as having possible contributions to Ciprofloxacin resistance.

# 1 Materials and methods

Throughout the paper we refer to fitness landscapes, and focus on fitness values as our phenotype of interest, but would like to note that the theory can be applied for any (binary) genotype-phenotype mapping and any phenotype. Let a fitness landscape be a mapping $f : \mathcal{P} \to \mathbb{R}$ from the genotype space to fitness values. Furthermore, suppose genotypes are strictly binary, $\mathcal{P} \in \{0, 1\}^p$, where 1 indicates the presence of a particular mutation (or variant), 0 indicates its absence, and $p$ is the number of genes. The complete fitness landscape then describes the effects of all combinations of mutations [16]. For example, the two gene space $\mathcal{P} = \{0, 1\}^2$ contains the wild-type 00, two single mutants 01 and 10, and the double mutant 11. The fitness landscape $f : \{0, 1\}^2 \to \mathbb{R}$ in this case can be written as

$$
\begin{aligned}
f(0,0) &= \beta_0 \\
f(1,0) &= \beta_0 + \beta_1 \\
f(0,1) &= \beta_0 + \beta_2 \\
f(1,1) &= \beta_0 + \beta_1 + \beta_2 + \beta_{1,2}
\end{aligned}
$$

for parameters $\beta_i \in \mathbb{R}$. $\beta_0$ is called the bias, $\beta_1$ and $\beta_2$ main effects, and $\beta_{1,2}$ the pairwise interaction. This pairwise interaction is exactly the classic definition of epistasis in quantitative genetics [17]. In generalising to higher-order interactions, we follow the definitions of [18]. For $p \geq 3$ genes, the complete fitness landscape $f$ is:

$$f(x_1, \ldots, x_p) = \beta_0 + \sum_i x_i \beta_i + \sum_{i<j} x_i x_j \beta_{i,j} + \sum_{i<j<k} x_i x_j x_k \beta_{i,j,k} + \ldots \tag{1}$$

While including all possible interactions quickly becomes computationally and statistically intractable for large $p$, we can model the interactions up to a point. Ignoring interactions of order four and higher we obtain an approximation of the fitness landscape:

$$f(x_1, \ldots, x_p) \approx \beta_0 + \sum_i x_i \beta_i + \sum_{i<j} x_i x_j \beta_{i,j} + \sum_{i<j<k} x_i x_j x_k \beta_{i,j,k} \tag{2}$$

The remainder of this section describes the regression model we use to estimate these effects, the algorithms we use to efficiently solve it, and finally the data sets on which we apply it.

### 1.1 Regression model

Given as input a matrix $\mathbf{X} \in \{0, 1\}^{n \times p}$ and a vector $\mathbf{Y} \in \mathbf{R}^n$, where columns of $\mathbf{X}$ are genes or variants of interest, rows are samples from the genotype space, and entries $y_i$ of $\mathbf{Y}$ are the fitness values corresponding to the $i$th row of $\mathbf{X}$, our goal is to estimate the parameters $\beta_0$, $\beta_i$, $\beta_{i,j}$, $\beta_{i,j,\ k}$ of the fitness landscape model in Eq (2), such that for any row $x_i$ of $\mathbf{X}$, $f(x_i) \approx y_i$.

To do this we construct a matrix $\mathbf{X}^* \in \{0, 1\}^{n \times p_{int}}$, where $p_{int} = \binom{p}{1} + \binom{p}{2} + \binom{p}{3}$, containing a column for each gene, pair of genes, and triplet of genes. Specifically, to construct $\mathbf{X}^*$ we extend $\mathbf{X}$ by adding the following columns. For every pair of columns $i, j$ and triplet $i, j, k$ we add an interaction columns $X_{i,j,k}$ by taking the element-wise product of the columns $X_i$ and $X_j$, or $X_i$, $X_j$, and $X_k$ (Fig 1).

Given the interaction matrix $\mathbf{X}^* \in \{0, 1\}^{n \times p_{int}}$ we estimate effects by solving the square-root lasso, as defined in [19], by minimising the difference between the predicted and actual fitness

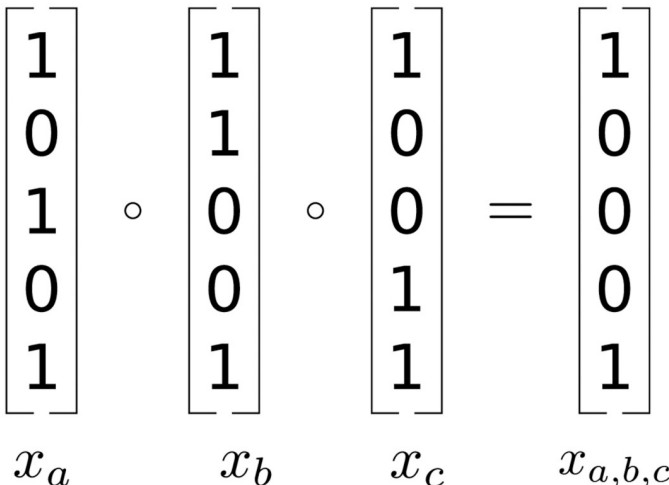

**Fig 1. Creation of three-way interaction effect columns.**

values, subject to regularisation.

$$\hat{\beta} = \underset{\beta \in \mathbf{R}^{P_{int}}}{\arg\min} \; \|y - \mathbf{X}^*\beta\|_2 + \lambda\|\beta\|_1 \tag{4}$$

## 1.2 Cyclic linear regression

Our approach to lasso regression is based on a cyclic coordinate descent algorithm from [20], as described in [8]. This method begins with $\beta_j = 0$ for all $j$ and updates the beta values sequentially, with each update attempting to minimize the current total error. Here this total error is the difference between the effects we have estimated and the fitness we observe. Where $y_i$ is the $i$th element of $\mathbf{Y}$, $\beta_j$ is the $j$th element of $\beta$, and $x_{ij}$ is the entry in the matrix $\mathbf{X}^*$ at column $j$ of row $i$, the error is the following:

$$\sum_{i=1}^{n}|r_i| \tag{3}$$

where the residuals, $r_i$, are the following:

$$r_i = y_i - \sum_{j=1}^{p_{int}} x_{ij} \cdot \beta_j \tag{4}$$

The error affected by a single beta value can then be minimized, using lasso regularisation, by updating $\beta_k$ as follows:

$$\beta_k \;\; \leftarrow \;\; \begin{cases} max(0, \beta_k + \dfrac{\sum_{i=1}^{n}(x_{ik}(y_i - r_i))}{|X_k|} - \lambda) & \text{for } \beta_k + \dfrac{\sum_{i=1}^{n}(x_{ik}(y_i - r_i))}{|X_k|} > 0 \\[3mm] min(0, \beta_k + \dfrac{\sum_{i=1}^{n}(x_{ik}(y_i - r_i))}{|X_k|} + \lambda) & \text{for } \beta_k + \dfrac{\sum_{i=1}^{n}(x_{ik}(y_i - r_i))}{|X_k|} < 0 \end{cases} \tag{5}$$

We adjust this to instead solve the square-root lasso ([21]) using Eq (6).

$$\beta_k \leftarrow \begin{cases} max(0, \beta_k + \dfrac{\sum_{i=1}^{n}(x_{ik}(y_i - r_i))}{E_{mse}} - \lambda) & \text{for } \beta_k + \dfrac{\sum_{i=1}^{n}(x_{ik}(y_i - r_i))}{E_{mse}} > 0 \\[3mm] min(0, \beta_k + \dfrac{\sum_{i=1}^{n}(x_{ik}(y_i - r_i))}{E_{mse}} + \lambda) & \text{for } \beta_k + \dfrac{\sum_{i=1}^{n}(x_{ik}(y_i - r_i))}{E_{mse}} < 0 \end{cases} \tag{6}$$

where

$$E_{mse} = \|Y - \mathbf{X}\beta\|_2$$

We cyclically update each $\beta_k$ until the solution converges for a particular $\lambda$, reduce the value of $\lambda$, and repeat. To reach the genome-scale we avoid unnecessarily considering most interactions (Section 1.4), compress the matrix (Section 1.5), and parallelise the process (Section 1.6).

## 1.3 Choosing lambda

The lasso penalty requires a regularisation parameter $\lambda$, which effectively decides how large an effect has to be before it will be included in the model. This can range from allowing all values ($\lambda = 0$) to allowing none ($\lambda \to \infty$). Choosing the correct value of $\lambda$ is essential if we want to include only the significant effects and avoid over-fitting noise. For the standard lasso this is typically done by choosing an initial value sufficiently large that all beta values will be zero and gradually reducing $\lambda$, fitting the model for each value until a stopping point chosen with

$K$-fold cross-validation [22]. Cross-validation requires fitting each $\lambda$ value $K$ times, however, significantly increasing the running time. The square root lasso performs well with an easily calculated $\lambda$ value, independent of the standard deviation of noise [21]. We use this lower limit of $\lambda = 1.1 \times \frac{1}{\sqrt{n}} \phi^{-1}\left(\frac{0.95}{2\times p}\right)$, where $\phi$ is the probability density function of the standard normal distribution and $p$ is the number of columns of the $\mathbf{X}^*$ matrix. Note that [21] actually use the penalty $\frac{\lambda}{n}$, whereas we use $\lambda$ directly as in [19]. To reach the same minimum value as in [21], our equation differs from theirs by a factor of $n$.

## 1.4 Pruning

We implement the branch pruning (Eq (7)) and working set (Algorithm 1) algorithms from WHInter [7] to avoid considering unnecessary effects at each value of $\lambda$.

The pruning algorithm determines whether any interaction with a particular effect $i$ can be non-zero at the current $\lambda$, and if not removes all such interactions $i, j$. Effects that may have a sufficiently large interaction are instead included in the working set. We give a brief overview of this algorithm here, and refer the reader to [7] for further details.

For the square-root lasso penalty we keep track of $\omega = \|r\|_2$ at each iteration, where $r = Y - \mathbf{X}\beta$. For a particular column index $x$ in $\mathbf{X}^*$, we further define:

$\rho_x$ = the residuals, the last time the column $X_x$ was included in the working set.

$\pi_x = \max\limits_{j \in (max(l), p]} \left(X_x^* \cdot X_j \cdot \rho_x\right)$

where $l$ is either the index of a single column of $\mathbf{X}$, or the set $\{i, j\}$ for the interaction column $X_x^* = X_i \circ X_j$. The upper bound $\eta(x)$ for any interaction with the column $X_x^*$ is then:

$$\eta(x) = \alpha \cdot \pi_x + \max\limits_{r \in \{r^+, r^-\}} \left|X_x^* \cdot r - \rho_x \cdot \alpha\right| \tag{7}$$

where:

$$r_i^+ = \begin{cases} r_i, & \text{if } r_i > 0. \\ 0, & \text{otherwise.} \end{cases}$$

$$r_i^- = \begin{cases} r_i, & \text{if } r_i < 0. \\ 0, & \text{otherwise.} \end{cases}$$

$$\alpha_x = \left|\frac{x \cdot r \cdot \rho_x}{\|\rho_x\|_2^2}\right|$$

According to the Karush-Kuhn-Tucker conditions for the square-root lasso, no effect can have a value less than $|\lambda \cdot \omega|$ [19]. We can therefore ignore any interactions whose effect has an upper bound below $|\lambda \cdot \omega|$. The working set contains the columns that have not been ruled out.

This fast step allows us to rule out most effects without even calculating them. Many interactions even among these columns will still never be updated at the current value of $\lambda$ however. As in [7] we further reduce the problem by calculating the *active set*, the subset of the working set that will be updated in the current iteration. To do so, we iterate through the entire working set one time, calculating all interactions and updating $\rho$ and $\pi$ for each, and adding those that are significant enough to the active set (Algorithms 1 and 3).

**1.4.1 Active-set for pairwise interactions.** For each pair $i, j$ of effects that have not been ruled out, we need the sum of their row's residuals $r \cdot X_i \cdot X_j$. As in [7], rather than taking all

the columns and reading through both to find the places they overlap we store the matrix in both column and row major versions and read through only the first column $\mathbf{X}_i$. All interactions are found by reading the row $k$ for each non-zero entry in $X_i$. Since the matrix is stored in a sparse format, we find all pairwise interactions with $\mathbf{X}_i$ in $O(\sum_{j=i+1}^{p} |X_i \circ X_j|)$ operations. This is further improved in our implementation by calculating a reduced row-major version of $X$, $X'$, containing only the effects present in the working set. Once the active set has been calculated, we solve the regression problem for the current $\lambda$ with Algorithm 2. In Pint we solve the main effects $\beta_i$ alone first, followed by pairwise effects $\beta_{i,j}$ (and finally three-way effects $\beta_{i,j,k}$). This ensures pairwise effects are only used to explain variance that cannot be fit using main effects. Similarly, three-way effects should only be introduced where pairwise effects are inadequate.

**Algorithm 1**: Identify Active Set (WHInter version, pairwise interactions only).

```
W ← ∅
for i ∈ 1 ... p do
  for a ∈ 1...n | X'_{a,i} ≠ 0 do
    sum_i ← sum_i + r_a;
    for j ∈ i+1...p | X'_{a,j} ≠ 0 do
      sum_{i,j} ← sum_{i,j} + r_a;
    end
  end
  γ ← |sum_i|;
  if sum_i > λ · ω then
    W ← W ∪ {{i}};
  end
  for j ∈ i + 1 ... p do
    γ ← max(γ, |sum_{i,j}|);
    if |sum_{i,j}| > λ · ω then
      W ← W ∪ {{i, j}};
    end
  end
  π_i ← γ;
  ρ_i ← r;
end
return W
```

**Algorithm 2**: Sequential Cyclic Sub-problem Algorithm.

```
while not converged do
  for k ∈ 1 ... p_{int} do
    Δβ_k ← (∑_{i=1}^n (x_{ik}(y_i − r_i))) / E_{mse};
    if |β_k + Δβ_k| > λ then
      β'_k ← β_k;
      β_k ← β_k + Δβ_k;
      if β_k > 0 then
        β_k ← β_k − λ;
      end
      else
        if β_k < 0 then
          β_k ← β_k + λ;
        end
      end
      for i ∈ 1 ... n do
        r_i ← r_i + x_{i,j} · (β_k − β'_k);
      end
    end
  end
end
```

**1.4.2 Active-set for three-way interactions.** WHInter's pruning algorithm (Algorithm 1) can be extended to three way interactions with a second-level pruning step while updating the active set (Algorithm 3). The three-way active set $W$ can then be solved as before using Algorithm 2. Since $\eta(x)$ requires the column $X_x^*$, to be calculated, and we are now using interaction columns $x = \{i, j\}$, we keep a cache of every interaction $X_{i,j}^*$ calculated so far and re-use them. For sufficiently small $\lambda$ this may become the majority of $\mathbf{X}^*$, so we compress these columns using Simple-8b (Section 1.5). The upper bound $\eta(\{i, j\})$ may be re-used and should also be cached.

**Algorithm 3**: Identify Active Set (Pint version, three-way interactions)

```
W ← ∅;
for i ∈ 1 … p do
  sum_i ← sum_i + r_a;
  for a ∈ 1..n | X'_{a,i} ≠ 0 do
    for j ∈ i + 1 … p | X'_{a,j} ≠ 0 do
      sum_{i,j} ← sum_{i,j} + r_a;
      if η({i, j}) > λ · ω then
        for k ∈ j + 1 … p | X'_{a,k} ≠ 0 do
          sum_{i,j,k} ← sum_{i,j,k} + r_a;
        end
      end
    end
  end
  γ ← |sum_i|;
  if sum_i > λ · ω then
    W ← W ∪ {{i}};
  end
  for j ∈ i + 1 … p do
    γ ← max(γ, |sum_{i,j}|);
    if sum_{i,j} > λ · ω then
      W ← W ∪ {{i, j}};
    end
    γ* ← 0;
    for k ∈ j + 1 … p do
      γ* ← max(γ*, |sum_{i,j,k}|);
      if sum_{i,j,k} > λ · ω then
        W ← W ∪ {{i, j, k}};
      end
      π_{i,j} ← γ*;
      ρ_{i,j} ← r*;
    end
    γ ← max(γ, γ*);
  end
  π_i ← γ;
  ρ_i ← r;
end
return W
```

## 1.5 Compression

In our method (Algorithm 3), the input matrix $\mathbf{X}$ is accessed frequently, iterating through both rows and columns. Because the matrix is not prohibitively large, we store it as both a column and row-major *uncompressed* sparse matrix. The active set, on the other hand, can be as large as $\mathbf{X}^*$. We considerably increase the number of possible non-zero effects by storing this only as a set of Simple-8b compressed columns (Fig 2(c)). Because we read the columns sequentially,

**Fig 2. Matrix compression.** Given a full matrix (a), we reduce it to the indices of non-zero entries (b), then the compressed difference between these (c). Arrows represent transitions between different representations.

we replace each entry with the offset from the previous entry. This reduces the average entry to a relatively small number, rather than the mean of the entire column. These small integers can then be efficiently compressed with any of a range of integer compression techniques (Fig 2), a subject that has been heavily developed for Information Retrieval. We compare a number of such methods, including the Simple-8b algorithm from [23] (which we implement and use in our package) in S1 Appendix.

## 1.6 Parallelisation

The three components of the algorithm, pruning, active set calculation, and solving the sub-problem, can all be done in parallel. Pruning and active-set calculation are trivially parallelisable, and performance scales well as long as each thread is given a large separate chunk of work. In practice this means dividing the matrix into several continuous chunks for the pruning step, one for each thread. For the active set calculation we calculate all two and three way interactions with a particular column on the same thread. As well as keeping each thread's workload sufficiently large, this means cached interaction columns and upper bounds can be kept thread-local.

The sub-problem (Algorithm 2) can also be parallelised, and performs well when the active set is sufficiently large and shuffled before each iteration [24]. Parallelising updates to a small active set can significantly harm performance however. In practice this presents a number of difficulties. The active set contains only the columns that would be non-zero at the current value of $\lambda$, and this is initially high, with the active set containing very few elements. It is therefore not worth parallelising until $\lambda$ reaches a value where sufficiently many non-zero effects are allowed. In our testing we often reached the final value, $\lambda = 1.1 \times \frac{1}{\sqrt{n}} \phi^{-1}\left(\frac{0.95}{2 \times p}\right)$, before this occurred.

Furthermore, as $\lambda$ decreases calculating the active set quickly dominates the running time. While not parallelising the sub-problem calculation theoretically limits the best-case performance of our method, it is a minor limitation in practice. We therefore keep this component single-threaded in our implementation in Pint.

We demonstrate the parallel scalability using a simulated data set of $n = 8,000$ rows and $p = 4,000$ columns, running Pint until the first 500 pairwise or main effects have been found and recording the median running-time out of five runs. As we see in Fig 3, performance scales

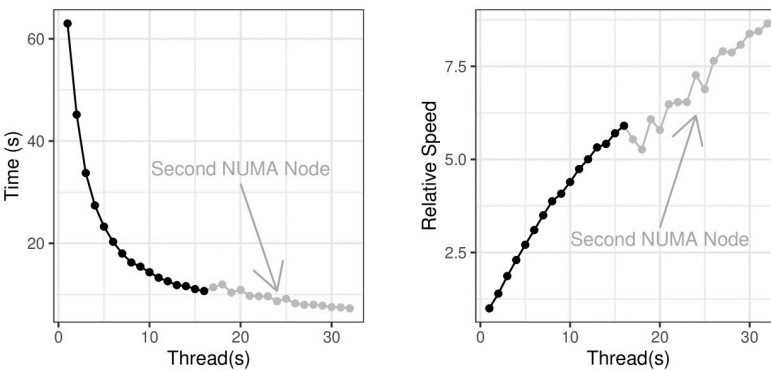

**Fig 3. Running time on an increasing number of threads.** Note that performance initially decreases with only a small number of threads on a second NUMA node. Tests were performed using two Intel Xeon Gold 6244 CPUs.

up to an 8× speed-up using 32 threads on 16 cores across two CPUs, which is typical for a highly parallel task running on CPU(s) with shared memory [25].

## 1.7 Approximate hierarchy

We include an approximate method to enforce a strong hierarchy by only allowing interactions between positions that have at some point had non-zero main effects. This is done by ignoring all interactions $i$, $j$, where one of $i$ or $j$ has a main effect strength of zero. In practice, this amounts to replacing the pruning step in *Section* 1.4 with one that simply includes main effects the first time they are assigned a non-zero value.

Doing so significantly reduces both the running time and memory use. We demonstrate this with a test case from the simulated data set, where $n = 4,000$, $p = 2,000$, containing 500 pairwise effects, each of which is composed of two main effects. Running to the lower limit for $\lambda$ without the approximate hierarchy constraint takes 42.5 seconds using eight SMT threads across four cores, with peak memory use of 6.87 GB. Adding the approximate hierarchy constraint, running time reduces to 9.98 seconds, and peak memory use to 2.32 GB. Since the running time and memory use depend on the number of main effects rather than $p$, these differences will only increase with larger values of $p$. The effect this has on accuracy is shown in Section 2.1.

## 1.8 Identifying identical columns

We include an option to ignore identical columns in the interaction matrix $\mathbf{X}^*$. These may be either direct columns of the input matrix $\mathbf{X}$ or interaction columns. We do so by computing the 128-bit hash of the column's non-zero entry positions using XXHASH [26]. All newly-considered columns have their hashes compared to those of previous columns, duplicates are placed on a list of known-duplicate columns and never included in the active set. This avoids spreading out an effect across multiple identical columns. Note that for non-binary matrices columns will be considered identical when the indices of their non-zero entries are the same, even if these entries differ.

## 1.9 Non-binary matrices

Real values may optionally be included in the matrix $\mathbf{X} \in \mathbb{R}^{n \times p}$, rather than strictly binary $\mathbf{X} \in \{0, 1\}^{n \times p}$, at the expense of running time. When real-value inputs are used, we maintain a

vector $V_k$ of the values for each column $\mathbf{X}_k$ of the input matrix $\mathbf{X}$. In working set calculations, we substitute $v_j x_{i,j}$, or $v_j v_k x_{i,l}$ where $l$ is an interaction between $j$ and $k$. In pruning we avoid actually calculating these values, instead we consider an upper bound on the possible interactions. We store the largest value in each column $k$ as $V_k^{max}$, and the largest value overall as $V_{all}^{max}$. The largest possible interaction with column $k$ is then:

$$\eta(x) = \alpha \cdot \pi_x + \max_{r \in \{r^+, r^-\}} ||V_{all}^{max}|| V_k^{max} | X_x^* \cdot r - \rho_x \cdot \alpha| \tag{8}$$

for pairwise interactions, substituting $|V_{all}^{max}|$ for $|V_{all}^{max}|^2$ in three-way interactions.

For large values of $V_{all}^{max}$ or $V_k^{max}$ this may include considerably more effects in the active set than if $\mathbf{X}$ were binary. In both the active-set calculation and the final regression step, we use the real value $v_j x_{i,j}$ in place of the binary value $x_{i,j}$.

## 1.10 Data

We prepared a collection of 70 simulated data sets and two experimental data sets to evaluate our method and test the scalability of our implementation. The first is the InfectX siRNA perturbation screen [27] in which siRNAs are applied to an infected human cell line. We predict off-target effects across the entire exome, and use these for our analysis. The second data set contains single nucleotide variants (SNVs) from 259 isolates of *Pseudomonas aeruginosa*, and associated minimum inhibitory concentration (MIC) of Ciprofloxacin.

**1.10.1 Simulated data.**    To evaluate the accuracy of our method Pint, we use benchmarks similar to [5]. To begin with, we simulate a matrix $\mathbf{X} \in \{0, 1\}^{n \times p}$ resembling siRNA off-target predictions for $n$ siRNAs across $p$ genes. We randomly assign effects to the silencing of some individual and pairwise combinations of the $p$ genes to produce effects $\beta_i$, and $\beta_{i,j}$. Our simulations differ from [5] in that we do nothing to ensure our pairwise effects are composed of existing main effects (i.e. we do not enforce a hierarchy). Taking the cumulative silencing effects $\Sigma_i X_i\beta_i + \Sigma_{i,j} X_i X_j \beta_{i,j}$, we add random noise from a normal distribution to produce a response vector $Y$, ensuring a signal-to-noise ration of 5. Each simulated is repeated several times to produce multiple data sets.

We simulate data sets of four different sizes. One with $n = 1,000$ siRNAs and $p = 100$ genes, repeated 50 times, a larger set with $n = 8,000$ siRNAs and $p = 4,000$ genes, repeated 10 times, and one with $n = 1,000$ siRNAs and $p = 10,000$ genes, repeated 10 times. Finally, we prepared a set including three-way interactions $\beta_{ijk}$, with $n = 400$ siRNAs and $p = 4000$ genes. The first represents an ideal scenario, with 10 siRNAs per gene at an easily tractable scale. The second is the largest set we are able to run with glinternet, and has only two siRNAs per gene. The third represents the worst case, where $p \gg n$. Each simulation in the $p = 100$ set contains 10 main effects and 50 pairwise effects. The $p = 4,000$ simulations contain 40 main effects and 200 pairwise effects. The wide $p = 10,000$ simulations contain 100 main and 500 pairwise effects. The three-way simulations contain 10 main, 100 pairwise, and 1,000 three-way effects.

We attempt to learn the gene silencing effects from the off-target matrix $\mathbf{X}$ and the response $Y$.

**1.10.2 InfectX siRNA data.**    To demonstrate our method on real genome-scale data, we use the mock group from InfectX [9]. This set contains 6,703 siRNA perturbations (excluding control wells and pooled siRNAs). Off-target effects are predicted using RIsearch2 [28], which includes a gene whenever there is a match between the siRNA seed region and some component of an mRNA for that gene (taken from [29]). We use an energy cut-off of −20 and match the entire siRNA (rather than only the 3′ UTR) as suggested in [28].

We then form the $6,703 \times 19,353$ matrix of off-target effects with columns for each gene, and rows for each siRNA as in [5]. An entry $i, j$ in this matrix is one if and only the predicted effect of siRNA $i$ on gene $j$ is greater than zero. All other entries are zero. Our fitness vector **Y** is the result of B-scoring then Z-scoring the number of cells in the well, to remove systematic within-plate effects and experimentally introduced cross-plate biases. B-scoring corrects for biases across the entire plate, Z-scoring then normalises each well's score with respect to the rest of it's plate.

**1.10.3 Antibacterial resistance.** SNVs from 259 isolates of *Pseudomonas aeruginosa* were sequenced using Illumina technologies (IPCD isolates on MiSeq and QIMR isolates on HISeq). SNVs from raw reads were mapped to the reference genome PAO1 using Bowtie2 (v. 2.3.4) [30] read aligners. Variant reports were then sorted into a table, set up so that each isolate was represented as a row and the presence / absence of each SNV was along the columns. Only genomes that had associated MIC values were included. We removed SNVs that occur less than five times, resulting in a table of 259 rows and 174, 334 columns.

*P. aeruginosa* genome sequences were selected from strains for which MIC values (Ciprofloxacin) were known. 167 genomes were sourced from the publicly available IPCD International Pseudomonas Consortium Database [31] and 92 genomes were from QIMR Brisbane Australia [32]. The IPCD data consisted of 2 x 300 bp MiSeq reads whilst the QIMR data was 2 x 150 bp reads. The MIC values were obtained as a combination of e-test strips [33] and plate-based assays [34, 35].

## 2 Results

In this section, we summarize the results of a simulation study we carried out to compare our method against existing approaches. We also demonstrate our method on two large-scale experimental data sets. We include these as reasonable examples of cases in which our method is applicable and validate the results by comparing them with known effects in the String and NCBI Gene databases [36, 37], as well the curated gene sets in the Molecular Signatures database [38].

### 2.1 Simulation performance

Our method aims to have comparable precision and recall to the best performing approach known to us [5] while scaling to much larger data sets. we compare precision and recall with glinternet, the most accurate of the methods tested in [5], and WHInter, a recent fast method based on the idea of limited working sets [7]. We use the simulated data described in Section 1.10.1, and consider only whether a method is able to correctly identify which effects are present, not whether the predicted effect strength is correct.

All methods use regularised regression in some form or another, and for a fair comparison we use the same parameters wherever possible. In each case we instruct the method to stop after the number of effects found significantly exceeds the number we simulated, $1,000$ in the small sets, and $5,000$ in the large sets and wide sets. We assume there is no bias $\beta_0$, and use a convergence threshold of 1%.

Although we give the same parameters, each method has a different halting criteria, and returns a different number of non-zero effects.

Pint typically predicts more non-zero effects than WHInter, but less than glinternet, the exceptions being the smallest ($p = 100$) data sets, where Pint predicts marginally fewer effects than WHInter, and the three-way interaction data sets, where Pint predicts more than either glinternet or WHInter.

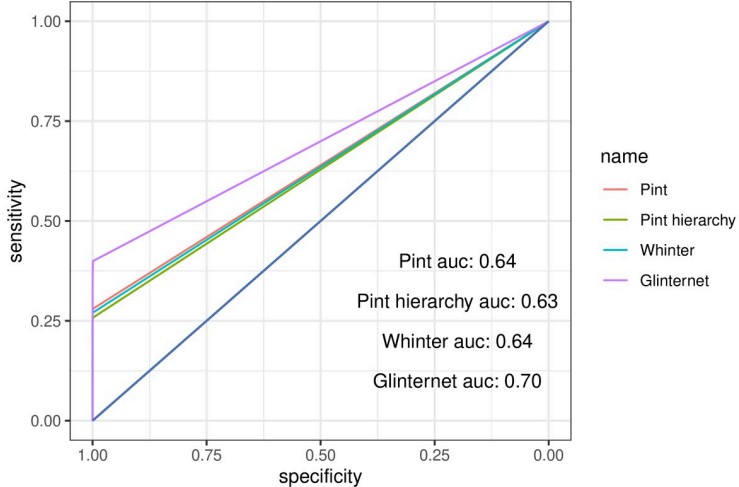

**Fig 4. Overall receiver operating characteristic curves for all $p = 4,000$ data sets.**

To keep the running time manageable we do not use cross-validation in glinternet. Pint and glinternet were run in parallel in the larger two benchmarks, using 48 SMT threads across 48 cores on two Intel Xeon 6342 CPUs. The smallest set is not large enough to benefit from parallelisation in Pint, and is run on a single thread. WHInter's implementation is single-threaded.

Comparing the ROC curves for each method, we typically find the highest area under the curve from glinternet, while Pint marginally outperforms WHInter (Fig 4 and Table 1). With all methods we see the predicted strongest effects tend to be correct, with true positives either identified as the strongest effects or not found at all (Fig 4).

The higher area under the curve for glinternet is likely a result of it's lower default regularisation threshold, allowing more non-zero effects. It should be noted that we did not use cross-validation, leaving glinternet to run until for either 200 iterations or until the minimum lambda of $\lambda_{min} = \frac{\lambda_{max}}{100}$ was reached, or the maximum allowed number of effects was found. Adjusting the lower lambda limit of Pint (Section 1.3) would increase the number of non-zero effects and might improve the ROC curve. Area under the ROC curve for all simulated data sets is shown in Table 1.

In every simulation Pint was significantly faster than both WHInter and glinternet (Fig 5 and Table 2). Typically, Pint was 5 to 6 times faster than WHInter, and approx. 90 to 1300 times faster than glinternet, when considering only pairwise effects (see Table 2). Adding the hierarchy assumption from Section 1.7 increases this to $\approx$ 50 times faster than WHInter in the wide test.

**Table 1. Overall AUROC for simulated data sets.**

|  | Pint | Pint (hierarchy) | WHInter | Glinternet |
|---|---|---|---|---|
| $p = 100, n = 1000$ | 0.747 | 0.720 | 0.744 | **0.780** |
| $p = 4,000, n = 8,000$ | 0.640 | 0.629 | 0.635 | **0.699** |
| $p = 10,000, n = 1000$ | 0.522 | 0.522 | 0.515 | **0.529** |
| $p = 400, n = 4,000$ (3-way) | **0.524** | 0.518 | 0.514 | 0.518 |

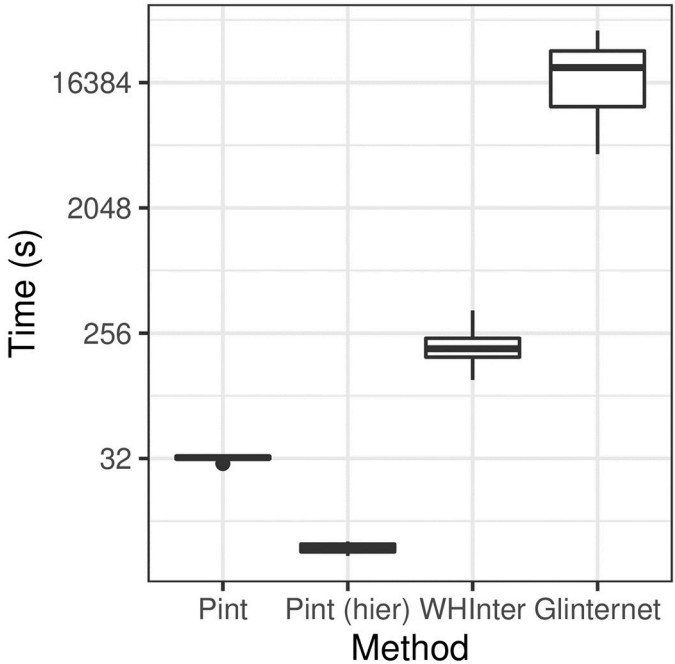

**Fig 5. Running times for each method using large $p$ = 4, 000 data sets.** Note that time is shown on a log scale.

We combine true positives and predictions from all repetitions to produce an overall ROC curve for each simulation, shown in Table 1. Note that while all methods performed poorly in the wide data sets, this is because there were a large number of effects to find, and regularisation kept most of these at zero. Among the non-zero effects, the strongest were still reliably true positives. Any of these methods could therefore be used with wide data, as long as the goal is only to determine a true subset of effects.

**2.1.1 Effect strength.** Among non-zero predictions, effect strength is a strong predictor of accuracy. While a significant number of true effects are not found with any method, the predicted strongest effects from each tend to be true positives. We see this in Fig 6, where only the non-zero predicted effects are included. Sorting by effect strength gives the majority of the true positives first for all methods, although Pint and WHInter significantly out-perform glinternet in this regard.

The ROC curves in Fig 6 imply that all methods, Pint and WHInter more so than glinternet, can achieve extremely high precision at the cost of poor recall, by considering only effects $i$ meeting a sufficient threshold of $|\beta_i|$. In practice this means that all methods can both be used to accurately find a subset of the true effects.

**Table 2. Mean time taken (s) for simulated data sets.**

|  | Pint | Pint (hierarchy) | WHInter | Glinternet |
|---|---|---|---|---|
| $p = 100, n = 1000$ | 0.08 | **0.06** | 0.40 | 86.3 |
| $p = 4, 000, n = 8, 000$ | 32.5 | **7.24** | 198 | 21019 |
| $p = 10, 000, n = 1000$ | 22.3 | **2.50** | 123 | 1888 |
| $p = 400, n = 4, 000$ (3-way) | 9.31 | **1.11** | 15.6 | 5632 |

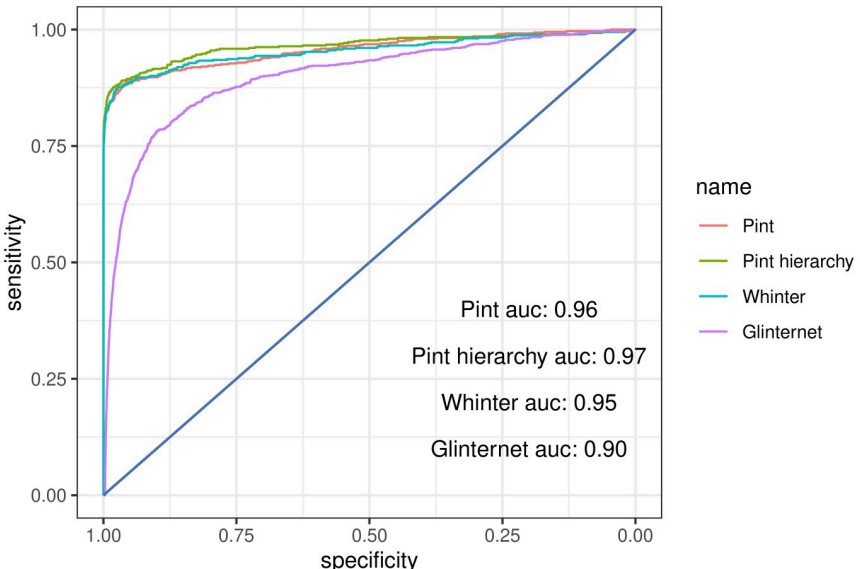

**Fig 6. Receiver operating characteristic curve comparing fraction of reported effects that are true positives as the predicted strength varies in the $p = 4,000$ simulations.**

## 2.2 Three-way effects

The majority of three-way effects were not assigned an effect with any method, resulting in poor overall performance (Fig 7 and Table 3). The inclusion of some correct three-way effects allows all varieties of Pint to outperform WHInter, however, and without the hierarchy assumption Pint also outperforms glinternet. This increases the running time, but Pint including three-way interactions is still faster than WHInter (Fig 7B).

Despite including three-way interactions, Pint does a poor job predicting them (Table 3). In particular, the vast majority of three-way interactions were not assigned any effect. Despite this, the strongest predictions tended to be correct, and considering only non-zero three-way predictions we have a AUROC of 0.85 and 0.84, with and without the hierarchy assumption respectively.

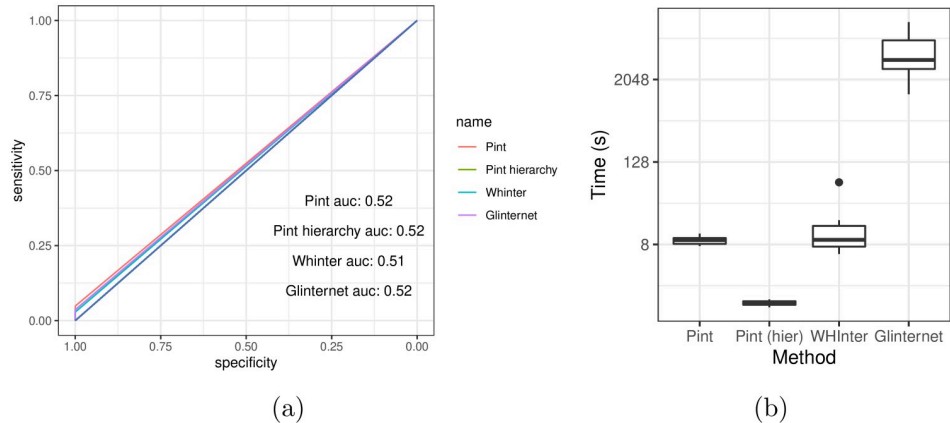

**Fig 7. Comparison of methods when 3-way interactions are present.** (a) ROC curve for full fit of all 3-way interaction data sets. (b) Time taken for each method. Note that Pint and Pint (hierarchy) are including three-way interactions, whereas glinternet and WHInter include only two-way.

**Table 3. AUROC for simulated three-way interaction data sets, separated by the type of effect.**

|           | Pint  | Pint (hierarchy) | WHInter | Glinternet |
|-----------|-------|------------------|---------|------------|
| Overall   | **0.524** | 0.518        | 0.514   | 0.518      |
| Main      | 0.850 | 0.843            | 0.830   | **0.867**  |
| Pairwise  | 0.613 | 0.606            | 0.615   | **0.643**  |
| Three-way | 0.511 | 0.504            | *NA*    | *NA*       |

## 2.3 InfectX siRNA data

We run our lasso model on the InfectX data (Section 1.10.2) allowing all pairwise and three-way interactions and halting at the end of the iteration once 100 non-zero effects are found. Only the combinations with non-zero predicted effects are then included in the matrix $\mathbf{Z}$. We then fit the fitness values $\mathbf{Y}$ to this matrix using least-squares regression $\mathbf{Y} \sim \mathbf{Z}\beta$, using these unbiased estimates and p-values as our final result. The resulting fit has an adjusted R-squared of 0.13 and an AIC value of 18, 184.82 We summarise the five most significant (according the p-value of the fit $\mathbf{Y} \sim \mathbf{Z}$) estimated effects in Table 4.

Among these effects, the three-way suppression of ANK1, KMT2D, and ZHX3 is particularly plausible. KMT2D is a known tumour suppressor, and mutations are common in lymphoma [39, 40]. ZHX3 is a transcriptional repressor, and in particular a failure of ZHX3 expression may be a cause of hepatocellular carcinoma [41]. Changes and failure to express KMT2D and ZHX3 respectively are associated with cancer development, and a significant increase in cell growth after suppressing both is consistent with these functions. ANK1 attaches integral membrane proteins to the cytoskeleton [42], and what role, if any, it plays is unclear.

The pairwise effect suppressing TRIM72 and ZNF264 could plausibly affect cell survival, as could suppressing PLCE1. PLCE1 is believed to play a role in cell survival and growth, and its suppression could have a significant effect on its own. It is unclear, however, how its suppression could have a positive effect on cell count [42]. TRIM72 plays a central role in cell membrane repair, and its suppression could easily affect fitness. ZNF264 may be involved in transcriptional regulation, and may have an interacting effect, although there are no known interactions between ZNF264 and TRIM72 [42].

The remaining two effects are TTC21A on its own, and RNF213 combined with SYN2. These genes are known to be involved in sperm function, vascular development, and neurotransmitter regulation respectively [42, 43]. We are not aware of any way in which RNF213 and SYN2 interact, or how either of these effects might affect cell growth or survival.

Taking the full set of 100 predicted effects, we search the Molecular Signatures Database [38] curated collection of gene sets. After correcting for multiple testing we find significant FDR-q values ($< 0.05$) for the following six sets:

**Table 4. InfectX most significant proposed effects.**

| Lasso Estimate | Least Squares Estimate | p-value      | Gene 1  | Gene 2 | Gene 3 |
|----------------|------------------------|--------------|---------|--------|--------|
| 0.16930307     | 0.1420643              | 0.0001837726 | PLCE1   | —      | —      |
| 0.31985354     | 0.2203255              | 0.0003231380 | ZNF264  | TRIM72 | —      |
| -0.09300632    | -0.1599196             | 0.0005853407 | TTC21A  | —      | —      |
| 0.14556737     | 0.1998505              | 0.0009281197 | ANK1    | KMT2D  | ZHX3   |
| -0.16530147    | -0.1874086             | 0.0010551688 | RNF213  | SYN2   | —      |

1. Genes up-regulated in the HMEC cells (primary mammary epithelium) upon expression of the transcriptionally active isoform of TP63 off adenoviral vector (q-value $2.42E - 05$).

2. Genes up-regulated in the HMEC cells (primary mammary epithelium) upon expression of TP53 off adenoviral vector (q-value 0.000582).

3. Expressed genes (FPKM > 1) associated with high-confidence PAX3-FOXO1 sites with enhancers in primary tumours and cell lines, restricted to those within topological domain boundaries (q-value 0.0025).

4. Genes up-regulated in HMEC cells (primary mammary epithelium) upon expression of both of TP53 and the transcriptionally active isoform of TP63 off adenoviral vectors (q-value 0.00778).

5. Genes that best predicted acute myeloid leukemia (AML) with the up-regulated expression of EVI1 (q-value 0.0247).

6. Top 40 genes from cluster 10 of acute myeloid leukemia (AML) expression profile, indicating poor survival (q-value 0.0482).

## 2.4 Antibacterial resistance

As explained in Section 1.10.3, our antibacterial data is pre-processed to remove variants present in less than five cases. The remaining 174, 334 columns are included in the model. We then fit the model $Y = X_2\beta + \epsilon$, where $Y$ is the MIC $log_2$ phenotype (indicative of Ciprofloxacin resistance). We include all pairwise interactions in $X_2$, and stop after 50 non-zero effects are found. These effects are included even with a large regularisation value λ, and are the most likely to be true positives (see Section 2.1.1). Again creating a $Z$ matrix with only the non-zero columns fitting $Y \sim Z$, we get a least-squares estimate with an Adjusted R-squared of 0.23.

The 50 non-zero effects involve 47 variants with 19 repeats and 16 pairwise effects (S1 Table). Of the pairwise effects, two pairs include the non-synonymous variant change that results in a Leu523Gln change in PA3054. PA3054 encodes a putative carboxypeptidase with the peptidase_M14 domain occurring between bases 24–634. Extracellular degradation of antimicrobials has been associated with increased production of M14 carboxypeptidases [44]. All other pairwise interactions identified involved synonymous variants. The most common of these was an A to G variant in PA3460, codon 537 Leu, found in 50% of the interactions. PA3460 encodes an acetyltransferase that is able to possibly modify Fluoroquinolones, reducing bacterial susceptibility to Ciprofloxacin [45]. The second most common synonymous variant was another A to G change in PA3709 encoding Ala 340 of a probable major facilitator superfamily (MFS) transporter protein. Overexpression of efflux pumps that include MFS transporters are associated with increased resistance to antibiotics [46, 47].

There were 34 variants that were characterised as contributing to Ciprofloxacin resistance. Of these, 16 were non-synonymous changes to proteins that are involved in fluoroquinolone modification, membrane transport or oxidative stress responses. The majority of non-synonymous variants occurred in oxidative stress response genes. An increase in reactive oxygen species (ROS) in response to Ciprofloxacin is well characterised in bacteria [48–50]. PA5401 encodes an electron transfer flavoprotein (EFT) domain-containing protein. The variant results in an Arg36Cys change in the β-subunit of an electron transfer protein whose gene is part of the dgc operon that is involved in choline metabolism and associated pathogenesis [51]. In eukaryotes, EFT is known to produce significant amounts of ROS in the presence of its partner enzyme medium-chain acyl-CoA dehydrogenase (MCAD) [52]. Therefore, non-

 

synonymous variants in PA5401 could result in changes to pathogenesis or ROS amounts. Our method also identifies effects for PA0117, pauD2, and gloA1, all of which are involved in glutathione production. Glutamine is a precursor of glutathione; glutamine and ascorbic acid have been found to provide substantial protection against Ciprofloxacin susceptibility in *Escherichia coli* [48].

Two genes with non-synonymous variants that were identified are involved in membrane integrity. The first is PA3173 with a His93Arg variation that encodes a short-chain dehydrogenase and acts on ubiG and ubiE involved in ubiquinone biosynthesis [53]. ROS accumulation affects membrane systems due to lipid peroxidation [54]. Ubiquinone is lipid-soluble and, therefore, is able to act as a mobile redox carrier within the cellular membrane [55]. Increased production of ubiquinone would reduce membrane damage caused by ROS. The second gene, MviN, with Leu316Met, is involved in peptidoglycan biosynthesis. Fosfomycin is frequently co-prescribed with Ciprofloxacin due to the synergistic activity of the two drugs [56]. However, increased peptidoglycan biosynthesis and cell wall recycling lead to antibiotic resistance [57]. Therefore the Leu316Met variant in MviN could be linked to increased resistance to combination therapy of Ciprofloxacin and Fosfomycin.

Overexpression of efflux pumps is a known contributor to increases in MICs for *P. aeruginosa*. Identified was a variant Lys329Gln in mexX that encodes a resistance-Nodulation-Cell Division (RND) multidrug efflux membrane fusion protein MexX precursor.

In total, 13 of the 16 non-synonymous variants have possible contributions to Ciprofloxacin resistance.

## 3 Discussion

Genotype-phenotype data sets have recently become available at a never before seen scale. In principle, it is possible to infer not only the effect of individual genomic variants within such data, but of pairwise and higher order combinations of their effects. While this has been shown to work in theory, and a number of tools have been developed that work on a smaller scale, there is a shortage of effective methods for human genome-scale data, and no method we are aware of includes three-way interactions. In this paper we present a lasso regression based method for such large-scale inference of pairwise and three-way effects.

Our method effectively performs coordinate descent square-root lasso regression on a matrix containing all pairwise and three-way combinations of the input data. We expand upon the method used in [7], with a number of improvements. We update the working set in parallel, resulting in a significant speed improvement. We extend the method from two-way interactions to three-way, adding pruning of pairwise effects. The active set is compressed with simple-8b, significantly reducing memory use and improving the running time with a large number of non-zero effects. We extend the method to include non-binary inputs in the **X** matrix, introduce an optional approximate hierarchy constraint that can be used to further reduce running time and memory use, and add detection of identical columns. Finally, we solve the square-root lasso instead of the lasso, giving us a well-defined stopping point.

We compared the accuracy and running time of our work to glinternet, the best of the methods we used previously [5] and WHInter, the fastest running method we are aware of [7]. Our simulations demonstrate comparable accuracy and recall to WHInter, while running approximately six times faster than WHInter and 600 times faster than glinternet in the largest tests. This goes up to 27 and $\approx 3,000$ times faster respectively when we assume interaction effects are hierarchical.

This performance improvement allows us to scale our method up to three-way interactions, while still running at the speed of WHInter with only pairwise interactions. In the presence of

    

three-way interactions, Pint is marginally more accurate than WHInter or glinternet. Assuming hierarchical interactions allows us to search for three-way interactions more quickly than either WHInter or glinternet find pairwise interactions. This comes at a slight cost to accuracy, however, which drops to match that of glinternet.

We also tested our method using two genome-scale real data sets. One is an exome-wide siRNA perturbation screen ($n \approx 6,700$ siRNAs and $p \approx 19,000$ genes). The other measures antibacterial resistance with respect to genetic variations in *Pseudomonas aeruginosa*, and includes over 15 billion possible pairwise interactions. In both cases our method finds a number of plausible interactions.

Despite this success, our method and its implementation in Pint have the following limitations. While our method is effective on genome scale data when using only pairwise interactions, running time limits the use of three-way interactions to smaller sets, or only the strongest interactions (running to completion was only possible with $p \leq 5,000$ in our testing). Furthermore, the sub-problem given the working set is not solved in parallel. While it is possible to do so, it is actually harmful to performance unless the working set is very large (i.e. many non-zero effects are included).

Note that while we consider only pairwise interactions Section 2.4, it is possible to include three-way effects if we remove columns with less than 30 entries instead. This reduces the input from 174, 334 to 75, 599 columns, and the first 50 interactions can then be found in approx. 80 hours. Alternatively, we could include three-way interactions by assuming a hierarchy, but we avoid doing so because it reduces overall accuracy in our simulations.

The additive interaction model is also an oversimplification of biology. It remains unclear to what extent genetic effects be treated as additive, and ignoring interactions among of more than three items could well be leaving out the most important effects. In this case we may end up spuriously associating phenotype changes with effects that just happen to be present, rather than the true, more complicated, interactions. Finally, we cannot distinguish interactions that are present in exactly the same rows of the input matrix. In the siRNA case, if two distinct pairs of genes are always suppressed by the same siRNAs, whichever is considered first will likely trump the other.

There are hence a number of opportunities to expand upon this work. While we can avoid including duplicate effects in our model (Section 1.8), we do not detect indistinguishable effects unless they are considered for inclusion in the working set. Moreover, we do not identify effects that are almost, but not quite, identical. Thoroughly accounting for the similarity of effects would further improve the model. Additionally, we chose the square-root lasso penalty partially because it has simpler to compute p-values than the lasso [19]. Implementing this would give unbiased p-values based directly on the lasso results, without requiring a second least squares fit. This is particularly important since the least squares p-values do not account for the column selection process, and are likely to be biased [58].

More generally, we could significantly increase the scale of interaction inference methods by reducing the search space. A more targeted approach estimating distance in 3D space using Hi-C [59] for example, would drastically reduce the time and space requirements, allowing higher order interactions to be considered. While we implement an optional approximate weak hierarchy constrain (Section 1.7), a strong hierarchy would further simplify the problem. It is worth noting however that these are not reasonable assumptions for all applications. Finally, the interactions proposed in Section 2.3 may be worth further investigation.

## Supporting information

**S1 Appendix. Compression methods.**
(PDF)

**S1 Table. Predicted top 50 SNV effects in the antibiotic data set.**
(PDF)

**S1 Fig. Compression effect on memory use.** Note that this is the total peak memory use of the program, not solely the memory used by the matrix $\mathbf{X}_2$. In both cases $n = 10 \cdot p$.
(TIF)

**S2 Fig. Comparison of compression methods.** (a) Total memory used, compressing the sparse $\mathbf{X}_2$ matrix with each method. (b) Total time taken and time taken (including compressing $\mathbf{X}_2$) and time taken for lasso regression alone, using each method.
(TIF)

## Author Contributions

**Conceptualization:** Kieran Elmes, Zhiyi Huang, Alex Gavryushkin.

**Data curation:** Kieran Elmes, Astra Heywood.

**Formal analysis:** Kieran Elmes.

**Funding acquisition:** Alex Gavryushkin.

**Investigation:** Kieran Elmes, Astra Heywood.

**Methodology:** Kieran Elmes.

**Resources:** Zhiyi Huang, Alex Gavryushkin.

**Software:** Kieran Elmes.

**Supervision:** Zhiyi Huang, Alex Gavryushkin.

**Validation:** Kieran Elmes.

**Visualization:** Kieran Elmes.

**Writing – original draft:** Kieran Elmes, Astra Heywood, Alex Gavryushkin.

**Writing – review & editing:** Kieran Elmes, Astra Heywood, Alex Gavryushkin.

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
