## [Decision Letter · Decision Letter 0]

8 Jun 2022

Dear Mr Elmes,

Thank you very much for submitting your manuscript "A fast lasso-based method for inferring higher-order interactions" for consideration at PLOS Computational Biology.

As with all papers reviewed by the journal, your manuscript was reviewed by members of the editorial board and by several independent reviewers. In light of the reviews (below this email), we would like to invite the resubmission of a significantly-revised version that takes into account the reviewers' comments. The reviewers found the proposed method to be sound and an improvement upon existing methods, but raised some concerns around the limitations and accuracy and precision of the results that need to be adequately addressed or justified.

We cannot make any decision about publication until we have seen the revised manuscript and your response to the reviewers' comments. Your revised manuscript is also likely to be sent to reviewers for further evaluation.

Sincerely,

Megan L. Matthews, Ph.D.

Guest Editor

PLOS Computational Biology

Ilya Ioshikhes

Deputy Editor

PLOS Computational Biology

Reviewer's Responses to Questions

**Comments to the Authors:**

Reviewer #1: The authors introduce a novel method based on lasso regression to identify features that have an effect on fitness. One novelty is to make the identification of order three effects feasible. They achieve this by improving a previous model used for order two interactions. Speed is improved through parallelization and memory reduced by data compression. Introduction of three-way effects is cleverly combined with the removal of uninformative pairwise effects a priori. The regression problem is solved by a square-root lasso.

The model seems mathematically sound and while the authors do not introduce new methodology, the implementation and combination of several approaches presents a novel software package useful for many relevant research topics, like for example drug resistance. However, I have some remarks that should be addressed.

I was able to install the package and run:

> output <- interaction_lasso(X, Y, n = dim(X)[1], p = dim(X)[2], lambda_min = -1, frac_overlap_allowed = -1, halt_error_diff=1.01, max_interaction_distance=-1, use_adaptive_calibration=FALSE, max_nz_beta=-1, max_lambdas=200, verbose=FALSE, log_filename="regression.log", depth=2, log_level="none", estimate_unbiased=FALSE, use_intercept=TRUE)

Error in interaction_lasso(X, Y, n = dim(X)[1], p = dim(X)[2], lambda_min = -1, :

unused arguments (frac_overlap_allowed = -1, use_adaptive_calibration = FALSE)

> output <- interaction_lasso(X, Y, n = dim(X)[1], p = dim(X)[2], lambda_min = -1, halt_error_diff=1.01, max_interaction_distance=-1, max_nz_beta=-1, max_lambdas=200, verbose=FALSE, log_filename="regression.log", depth=2, log_level="none", estimate_unbiased=FALSE, use_intercept=TRUE)

total entries: 49944

This should obviously be fixed in the readme of the package.

github.com/bioDS/lasso_data_processing leads to a 404.

p.10:

I looked at reference [15] to understand how the simulated data sets were constructed. However, the authors should at least address the sizes of the data sets directly in their manuscript’s main part. This is important information and should be readily available to the reader. If I understand correctly, the sizes are nxp = 1000x100 and 10000x1000. That seems relatively small compared to the real data sets with dimensions 6703x19533 and 259x174334. The authors should explain why the relation of n and p in the real data sets is inverse to the simulated ones. In the discussion the authors mention /approx 67000 siRNAs and the n=6703 from before could be a mistake. The authors address the limitation, where they talk about data set size and three-way effects, i.e., when their method breaks down. Why is that not directly shown in the simulations? The authors should add more simulations to roughly show when this three-way effect identification becomes infeasible.

What is in general very concerning is the overall low accuracy of all methods, even for small data sets. The authors need to explain why such low precision and relatively high (only for small data sets) recall is favored in their performance. Are false positives considered the lesser evil? The authors should consider the area under the curve for the ranked effects, i.e., precision and recall for the first (i.e., strongest) effect that is found, then for the first and second, then for the first, second and third effect, and so on. This might even be a fairer evaluation of usefulness and lets us know if false positives and/or false negatives rank very low or high (which would be concerning). Additionally, the authors never show recall for the 3-way effects. The question is, how relevant is this with a precision of 6%? How good is random in this case? I.e., what is the prevalence of the effects in the data sets? If I read [15] correctly, the data sets include only {5, 20, 50, 100} + {0, 20, 50, 100} main and pairwise effects. What about three-way effects in the simulated data? That is of course never talked about in [15] but also not in the main manuscript. Hence, the referral to the simulated data in [15] makes no sense for three-way effects.

Just to be clear, the authors never use the actual effect size for performance evaluation. The main concern is just to identify true positives, correct? Maybe they should make that a bit clearer in the text.

As I understand, the authors look for cell growth and survival genes in their list of identified effects from the infectx data. While the top found genes seem to certainly make sense and are a nice result in itself, I suggest if possible an additional gene set enrichment analysis to make this part more systematic and quantifiable. This option does not seem to be that straightforward for the second data set, does it? "... so the possibility of a true combined effect of Ciprofloxacin resistance cannot be ruled out... ", but seems highly unlikely given the simulation results. Also, even if the variants were characterized as not affecting Ciprofloxacin resistance, even then a true combined effect could still not be ruled out. Hence, this last part of the sentence seems like a NULL statement and should be removed, unless I am completely missing the meaning of that sentence. E.g., what are "majority variants"? Shouldn't that be "majority of variants"? I noticed several long sentences like this one and suggest in general to keep sentence length as concise as possible..

p.13:

"Additional genes identified using our method...". Did you use any other methods to identify genes in this section? If so, this is not clear at all.

The authors again make the argument in the discussion that tools need to be tailored towards larger data sets and higher order interactions. That is why they developed Pint. However, if accuracy is so low for relatively small data sets (where speed is not the issue), how can they make any prediction on even larger data sets? Accuracy will drop even more. Like mentioned above maybe the top hits still look very good and an AUC as accuracy measure might visualize that in the simulations.

The authors mix notation, which can be confusing. E.g., /phi is used once to denote the probability density function of the normal distribution and another time as "r, the last time column x was included in the working set" (p.5). This type of ambiguity should be avoided. It is also confusing reading about column X_x and column x, which is also referred to as the index ("... interaction with x ...", this time x is the column? p.5). The authors should make sure that their notation is consistent and unambiguous throughout the whole manuscript. Also defining /phi_x = r and then using r/cdot/phi_x in an equation seems strange. It would be less confusing just defining /phi_x as "the residuals, the last time column x was inc..." without misusing r. It seems that the residuals are used on page 4 eq.4 for the first time. Even though it is a common notation in the field, they still need to be formally introduced.

p.9:

Off-target effects are prediction using RIsearch2 => Off-target effects are predicted using RIsearch2

In the follow up sentence: the gene should be called the "off-target". The respective off-target effect would be the inhibition of the gene's expression.

p.10:

"... further increasing the number of threads has to noticeable impact on performance..." => "... further increasing the number of threads has no noticeable impact on performance..."

p.11:

The "Method" in figure 4 is supposed to be Pint, right? Please correct that/make that clear.

p.16:

The reference of [38] seems to be incomplete.

P.19:

“We can considerably reduce the size of the active set by compression the columns.” => “We can considerably reduce the size of the active set by compressing the columns.”

Reviewer #2: In this paper Elmes et al proposed a fast LASSO-based method, Pint, for inferring higher-order interactions. This method was claimed to outperform known methods in simulated data, and identifies a number of biologically plausible gene effects in both

the antibiotic and siRNA models. This research was motivated in large genetic/genomic studies where a large number of genetic/genomic features are present and their interactions need to be assessed. In such studies because of the number of features (include interactions), the computing efficiency or scalability becomes a main issue. One major innovation of Pint compared to the existing approaches such as glinternet and WHinter is it’s better computing efficiency achieved in implementation including parallelization. This efficiency is very much needed in such studies. The package was shown to discover interesting three-way interaction terms in the siRNA data. This paper is well written, the methods are sound and results are convincing. It presents significant development in implementing square-root LASSO method. It will be of interest of general audience in this field. I recommend to publish in your journal.

I have a few comments and would like authors to address:

1. In Figure 4c, it’s noted that the precision for the 3way interaction terms was extremely low. In Section 3.2 the authors noted that this was mostly due to overlapping three-way effects? Could authors clarify what they meant by “overlapping three-way effects”? columns are identical? If so, how many False positives due to such overlapping three-way effects? When n=1000 in the binary setting, the interaction term is either 0 and 1. How many such three way interaction terms were detected significant? Such low precision may limits the usefulness of the method in such applications, which may deserve some discussion

2. Can Pint apply to non-binary data? I can see many applications may go beyond the SNP type of variables, (ie. continuous instead of binary). But the binary setting of variables may greatly limit the usage of this method. For example, in gene-gene interaction case, all expression may be continuous. The overlapping-effect presented in Figure 4C maybe not an issue. If Pint works for such data, it will expand the scope of applications. Can authors discuss it?

Minor point:

On page 7, section 2.1 should X\\in \\{0,1\\}^p be X\\in \\{0,1\\}^{nxp}?

**Have the authors made all data and (if applicable) computational code underlying the findings in their manuscript fully available?**

Reviewer #1: **No: **github.com/bioDS/lasso_data_processing leads to a 404.

Reviewer #2: Yes

PLOS authors have the option to publish the peer review history of their article (what does this mean?). If published, this will include your full peer review and any attached files.

Reviewer #1: No

Reviewer #2: No
---

## [Decision Letter · Decision Letter 1]

24 Sep 2022

Dear Mr Elmes,

Thank you very much for submitting your manuscript "A fast lasso-based method for inferring higher-order interactions" for consideration at PLOS Computational Biology. As with all papers reviewed by the journal, your manuscript was reviewed by members of the editorial board and by several independent reviewers. The reviewers appreciated the attention to an important topic. Based on the reviews, we are likely to accept this manuscript for publication, providing that you modify the manuscript according to the review recommendations. Particularly, please address the additional comments from Reviewer 1 about the AUC and PR plots and the enrichment analysis.

Sincerely,

Megan L. Matthews, Ph.D.

Guest Editor

PLOS Computational Biology

Ilya Ioshikhes

Section Editor

PLOS Computational Biology

[LINK]

Reviewer's Responses to Questions

**Comments to the Authors:**

Reviewer #1: I appreciate the authors’ effort to address all of my previous points. However, some issues remain, which to me seem fixable with relatively little work and would significantly increase the quality of the manuscript.

Maybe the authors misunderstood my previous concerns regarding the precision/recall plots and my suggestion to replace (not add) them with AUC plots. Precision and recall in the single digits do not show that a method is working reasonably well (on the contrary). However, precision and recall are also much dependent on the cutoff for the output values (in this case !=0 and ==0, correct?). Considering a small effect only marginally larger than zero as an effect and a large effect far larger than zero as the exact same effect seems unreasonable. The AUC does not depend on such a cutoff. Hence, I strongly suggest that the authors replace precision/recall plots with AUC plots unless there is a valid reason not to do so (which I can not see). The single shown AUC plot seems to show that the method works very well (is there a corresponding precision/recall plot to it?, because I have a hard time believing that this AUC plot would result in low recall/precision numbers or is this simulation run an outlier? There is no error shown.). Especially, since the authors let their method stop after some (arbitrary? At least in the application to real data. How is that number determined?) number of found non-zero effects, the AUC shows that the top effects should be true positives.

I do not follow the explanation of why an enrichment analysis is not possible. You have your identified genes and there are databases (e.g., http://www.gsea-msigdb.org/gsea/msigdb/index.jsp) with known biological processes and other common functional gene groups available. Based on this you could for example compute a simple confusion matrix (overlap of your genes and the group, exclusive to the group/your genes, and rest of the gene population) and use Fisher’s exact test (or equivalent) to try to quantify the overlap. There is no need to rank the results.

Other:

I assume the authors actually simulated each data set several times to get the variance of the accuracy plots. Is that and how often this is done stated anywhere?

“... we maintain a vector Vk of the values for each column of the input matrix Xk.” => “... we maintain a vector Vk of the values for each column Xk of the input matrix X.”

“... We prepared one simulated and two experimental data sets …” Should be nine (without repetition for random noise) simulated data sets, right?

“To begin with, we take simulated a simulated matrix …” => “To begin with, we simulated a matrix …”

“datasets” => “data sets” (or be consistent with datasets).

Response: “In summary, yes for the applications we had in mind the main objective was to identify true positives. “

But this objective is not fulfilled, if you identify a lot of false positives due to low precision. In a real case you cannot distinguish true and false positives. I fail to see the motivation here.

Reviewer #2: The authors have addressed my comments

**Have the authors made all data and (if applicable) computational code underlying the findings in their manuscript fully available?**

Reviewer #1: Yes

Reviewer #2: Yes

PLOS authors have the option to publish the peer review history of their article (what does this mean?). If published, this will include your full peer review and any attached files.

Reviewer #1: No

Reviewer #2: No

Figure Files:

Data Requirements:

Reproducibility:

References:

---

## [Decision Letter · Decision Letter 2]

11 Nov 2022

Dear Mr Elmes,

We are pleased to inform you that your manuscript 'A fast lasso-based method for inferring higher-order interactions' has been provisionally accepted for publication in PLOS Computational Biology.

Best regards,

Megan L. Matthews, Ph.D.

Guest Editor

PLOS Computational Biology

Ilya Ioshikhes

Section Editor

PLOS Computational Biology

Reviewer's Responses to Questions

**Comments to the Authors:**

Reviewer #1: I greatly appreciate the additional amount of work the authors put in and think all my points have been completely satisfied.

**Have the authors made all data and (if applicable) computational code underlying the findings in their manuscript fully available?**

Reviewer #1: Yes

PLOS authors have the option to publish the peer review history of their article (what does this mean?). If published, this will include your full peer review and any attached files.

Reviewer #1: No

---

## [Editor Report · Acceptance letter]

9 Dec 2022

PCOMPBIOL-D-22-00040R2 

A fast lasso-based method for inferring higher-order interactions

Dear Dr Gavryushkin,

I am pleased to inform you that your manuscript has been formally accepted for publication in PLOS Computational Biology. Your manuscript is now with our production department and you will be notified of the publication date in due course.

With kind regards,

Zsofi Zombor
